# Mean-based Best Arm Identification in Stochastic Bandits under Reward Contamination

**Arpan Mukherjee**
Rensselaer Polytechnic Institute
mukhea5@rpi.edu

**Ali Tajer**
Rensselaer Polytechnic Institute
tajera@rpi.edu

**Pin-Yu Chen**
IBM Research
Pin-Yu.Chen@ibm.com

**Payel Das**
IBM Research
daspa@us.ibm.com

## Abstract

This paper investigates the problem of best arm identification in *contaminated* stochastic multi-arm bandits. In this setting, the rewards obtained from any arm are replaced by samples from an adversarial model with probability $\varepsilon$. A fixed confidence (infinite-horizon) setting is considered, where the goal of the learner is to identify the arm with the largest mean. Owing to the adversarial contamination of the rewards, each arm's mean is only partially identifiable. This paper proposes two algorithms, a gap-based algorithm and one based on the successive elimination, for best arm identification in sub-Gaussian bandits. These algorithms involve mean estimates that achieve the optimal error guarantee on the deviation of the true mean from the estimate asymptotically. Furthermore, these algorithms asymptotically achieve the optimal sample complexity. Specifically, for the gap-based algorithm, the sample complexity is asymptotically optimal up to constant factors, while for the successive elimination-based algorithm, it is optimal up to logarithmic factors. Finally, numerical experiments are provided to illustrate the gains of the algorithms compared to the existing baselines.

## 1 Introduction

**Overview.** This paper investigates the problem of *best arm identification* (BAI) in stochastic multi-armed bandits (MABs), under the key assumption that the reward samples are vulnerable to adversarial corruptions. Specifically, consider a $K$-armed stochastic MAB, where the reward from arm $i \in [K] \triangleq \{1, \cdots, K\}$ is drawn from a sub-Gaussian distribution $\mathbb{P}_i$. At each round $t$, the observed reward may be corrupted with probability $\varepsilon$. When a reward sample is corrupted at time $t$, it is replaced by a sample drawn from a contamination model with distribution $\mathbb{Q}_i(t)$, which is distinct from $\mathbb{P}_i$. Such a setup is a non-trivial generalization of the canonical BAI problem due to the arbitrary corruption that may be introduced by the corruption model. This setup has far superior practical implications. For instance, consider the example of measuring drug responses, in which several compounds are evaluated in order to determine which one of them renders the maximal efficacy. It is natural to assume that a fraction of the test results are reported incorrectly, or a fraction of the samples may be contaminated [1]. Both possibilities affect the measurements (or rewards) that are used to identify the most effective drug. In a different example, consider a recommendation system in which the goal is to recommend the most "interesting" articles to users based on their feedback on previous recommendations. In this case, likewise, user feedbacks are prone to be imprecise or even malicious in a fraction of the recommendations. Such examples motivate investigating the BAI problem under the additional consideration that there exists the possibility that a fraction of the reward samples are

35th Conference on Neural Information Processing Systems (NeurIPS 2021).

corrupted and they obscure the process of identifying the true best arm. The problem of contaminated best arm identification (CBAI) has been studied under two main settings, namely the fixed-budget setting and the fixed-confidence setting. The objective in the fixed-budget setting is to identify the best arm within a given sampling budget such that the misclassification rate is minimized [2]. On the other hand, that of the fixed-confidence setting is to obtain a prescribed confidence with the fewest samples possible, on average [3]. In this paper, we investigate CBAI in the second setting, in which we are prescribed a fixed guarantee on the misclassification probability and the goal is to identify the arm with the highest true mean, while, in parallel, minimizing the sample complexity. The main contributions of this paper are the following.

- This paper is the first to analyze the CBAI problem for sub-Gaussian bandits in the fixed confidence setting, without altering the canonical definition of the best arm which is defined as the arm with the highest mean reward. The existing literature on CBAI aims to identify arms with the largest median [4], which is distinct from the definition of the best arm in the canonical BAI problem.

- We provide two algorithms to address the CBAI problem, one of them being a procedure based on successive elimination, and the other being a confidence gap-based procedure. Both algorithms use the trimmed mean as an estimator. We provide closed-form decision rules for dynamically selecting the arms over time, estimating the means corresponding to every arm of the bandit instance, stopping the arm-selection process, and finally identifying the true best arm.

- We analyze the attendant performance guarantees of the proposed algorithms. Specifically, we establish the decision reliability and the sample complexity. While the algorithm based on successive elimination is shown to be optimal up to logarithmic factors, the one based on confidence gap is shown to be asymptotically optimal up to constant factors. Both algorithms can identify the best arm even when a fraction of the reward samples are corrupted.

- Finally, we conduct numerical simulations to demonstrate the efficacy of the proposed algorithms. The experiments show that both algorithms outperform the existing methods for CBAI in both synthetic as well as real-world datasets (content recommendation and drug testing).

**Related Literature.** We review the existing literature in two parts – first, the literature on BAI in the fixed-confidence setting, and next, the literature on contaminated stochastic MABs. The problem of BAI for stochastic MABs has a rich literature, dating back to the study in [5]. Specifically, the existing BAI algorithms can be broadly grouped into two categories: (i) algorithms that are guided by confidence bounds, and (ii) those that involve successively distilling the search space. The former class of algorithms involve constructing confidence sets that capture the deviation of the empirical mean of an arm from its true mean. These algorithms terminate and recommend a predicted best arm once the accumulated data is sufficient for forming a decision that meets a prescribed confidence level. Some of the representative studies in this class include [6], [7], [8], and [9]. The other category of algorithms, also known as racing algorithms, involve successively identifying and rejecting the suboptimal arms until only one arm is left in the active arms set. One of the first of these algorithms was proposed in [10]. Other examples of racing algorithms for stochastic MABs include the studies in [8], [11], [12], and [13].

The literature on contaminated stochastic MABs includes analyzing adversarial corruptions in stochastic MABs for regret minimization, first studied in [14]. In this study, the adversarial power is characterized by the total number of corrupted samples $C$ that can be introduced by the adversary until the specified time horizon. The algorithm proposed by this study degrades linearly with $C$, which is the optimal rate for any algorithm that achieves the optimal performance in the stochastic setting. The regret bound obtained in this study is further improved in [15]. More recent studies in this direction include [16, 17, 18, 19, 20]. The problem, however, is less-investigated in the context of BAI. In the fixed-budget setting, the problem of CBAI is studied in [21]. This study proposes a probabilistic sequential algorithm, which allows for trading off the suboptimality gap (in the fully stochastic regime) with the success probability using a tunable parameter. In the fixed-confidence setting, CBAI is studied in [4], which modifies the definition of the "best arm" by redefining it to be the arm that has the largest *median*. A general adversarial model is assumed in this study, where any reward sample can be contaminated by the adversary with probability $\varepsilon$. Furthermore, three different attack models are considered, namely, the oblivious adversary, the prescient adversary, and the malicious adversary, depending on the level of adaptivity that the adversary employs to obfuscate the true reward samples. While the investigation proposes instance-adaptive algorithms matching up to logarithmic factors, in many models, the median and the mean could be considerably different,

especially for the case of heavy-tailed distributions. Furthermore, the study considers a considerably restrictive class of statistical models (Definition 2, [4]), which does not include several popular and widely-analyzed classes of bandit instances (e.g., heavy-tailed continuous models and all discrete models such as the class of Bernoulli bandits). In this paper, we investigate these questions under the conventional definition of the best arm, where the best arm is defined as the arm having the largest *mean*. Furthermore, we investigate the entire class of sub-Gaussian bandits.

## 2 Contaminated best arm identification

**Setting.** Consider the canonical bandit model $\nu \in \mathsf{SG}_K(\sigma)$, where $\mathsf{SG}_K(\sigma)$ denotes the class of $K$-armed $\sigma$-sub-Gaussian bandits. The reward of arm $i \in [K]$ is generated from a probability measure $\mathbb{P}_i$ with mean $\mu_i \in \mathbb{R}$. The means $\{\mu_i : i \in [K]\}$ are assumed to be unknown. At each time $t$, a learner interacts with the bandit instance by pulling one of the arms $A_t \in [K]$ according to a control policy, generating the random reward $X_{A_t,t} \in \mathbb{R}$. Based on the observed reward, a decision is made about the next arm to be pulled. Additionally, the setting assumes an adversary that is capable of contaminating the reward before it is observed by the learner. Specifically, it is assumed that at each time $t$, the adversary flips a coin, whose outcome is denoted by the random variable $D_t$, where $D_t \sim \mathsf{Bern}(\varepsilon)$. Depending on the outcome of the coin toss, the adversary sends the true reward sample, or an adversarial sample drawn from a corruption model. Thus, with a fixed probability $\varepsilon \in (0,1)$, the adversary replaces the true reward with a corrupt sample drawn from an adversarial distribution $\mathbb{Q}_i(t)$ distinct from $\mathbb{P}_i$. We consider an *oblivious* adversary which is responsible for the contamination of the reward samples. Let us define $X_{i,t}$ as the reward generated by arm $i \in [K]$ at time $t$ according to the true distribution $\mathbb{P}_i$. The adversarial samples for each arm $i \in [K]$ at each time instant $t$ is denoted by $X'_{i,t}$. An oblivious adversary is defined as follows.

**Definition 2.1** (Oblivious adversary). *An adversary is said to be oblivious if for all $i \in [K]$, the sequence of triples $(X_{i,t}, X'_{i,t}, D_t)_{t \geq 1}$ are assumed to be independent.*

Hence, at each time $t$, the adversarial action can be characterized by a Bernoulli random variable $D_t \sim \mathsf{Bern}(\varepsilon)$, based on which the observed reward takes the form

$$R_{A_t} = \begin{cases} X_{A_t,t} \sim \mathbb{P}_{A_t}, & \text{if } D_t = 0 \\ X'_{A_t,t} \sim \mathbb{Q}_{A_t}(t), & \text{if } D_t = 1 \end{cases} . \tag{1}$$

This contamination model was first proposed in [22] in the canonical estimation framework under a single control action. Thus, the contaminated model corresponding to each arm of the bandit instance is characterized by the mixture distribution

$$\tilde{\mathbb{P}}_i(t) = (1 - \varepsilon)\mathbb{P}_i + \varepsilon\mathbb{Q}_i(t) . \tag{2}$$

The learner's sequence of actions and rewards are denoted by

$$\mathbf{A}^t \triangleq [A_1, \cdots, A_t] \qquad \text{and} \qquad \mathbf{R}^t \triangleq [R_{A_1}, \cdots, R_{A_t}] . \tag{3}$$

**Partially Identifiable Best Arm Identification (PIBAI).** Based on the sequence of rewards accumulated over time, the goal of the learner is to identify the best arm with a high probability. Let us denote the index of the best and second-best arms of the bandit instance by $a^\star$ and $b^\star$, respectively. Due to the action of the adversary, it is impossible to identify the true mean of any arm $i \in [K]$, even with an infinite number of samples from that arm [4]. Hence, we consider the notion of *partially identifiable best arm identification* (PIBAI). Let us consider arm $i \in [K]$ of the bandit instance $\nu$. Under the PIBAI setting, we can only estimate the mean of an arm up to a constant uncertainty interval $U_i$ around the true mean, i.e., the interval $\mathcal{I}_i \triangleq [\mu_i - U_i, \mu_i + U_i]$. Subsequently, in the context of PIBAI, the goal of identifying the best arm can be presented using the following guarantee on the identification performance.

**Definition 2.2** ($\delta$-PAC). *In the context of PIBAI, for a given set of uncertainties $\{U_i : i \in [K]\}$, we say that a procedure is $\delta$-PAC if it satisfies the following guarantee.*

$$\mathbb{P}\Big(\mu_{\hat{a}_\tau} + U_{\hat{a}_\tau} < \mu_{a^\star} - U_{a^\star}\Big) \leq \delta , \tag{4}$$

*where $\tau$ denotes the stopping time of the procedure.*

We note that CBAI is a special case of the PIBAI framework. Specifically, for a fixed level of contamination $\varepsilon$, the bias in estimation is the same for each arm $i \in [K]$, that is, $U_i = U$ for every $i \in [K]$. Here, $U$ is a function of the level of contamination $\varepsilon$ and the variance $\sigma$ of the arms [23].

**Key assumptions.** Now, we formalize the key technical assumptions that are necessary for the algorithms proposed in Section 3. The first assumption ensures the feasibility of being able to identify the true best arm. Essentially, there always exists adversarial models that could render the goal of CBAI infeasible, even if we have an infinite number of samples. We assume that the algorithms operate in a regime in which it is possible to identify the true best arm. The next assumption is a standard assumption in the robust statistics literature, and it provides an upper bound on the maximum level of corruption that the adversary can inflict up on the rewards. The assumptions are formally stated below.

    (i) The index of the best arm does not change as a result of contamination, i.e., $(\mu_{a^\star} - U_{a^\star}) > (\mu_a + U_a)$ for every $a \in [K] \setminus a^\star$.

    (ii) We do not require to know the precise value of the level of contamination $\varepsilon$. Rather, it is sufficient to assume that we know an upper bound on it, such that at each time $t$, the fraction of contaminated samples falls below the upper bound. Henceforth, for the rest of the paper, $\varepsilon$ denotes an upper bound on the probability of adversarial replacement of rewards. We focus on the regime $\varepsilon < 1/2$.

## 3 CBAI algorithms

We provide two CBAI algorithms in this section, and the attendant performance guarantees will be analyzed in Section 4. Specifically, we propose two instance adaptive algorithms, one based on the principle of successive elimination of suboptimal arms, inspired by [24], and the other one is a gap-based algorithm, which aims to reduce the overlap among the confidence intervals for different arms. One common method shared by both the algorithms is the estimator for the largest mean. We discuss this shared procedure before discussing the two algorithms.

---

**Algorithm 1** Gap-based algorithm for CBAI (G-CBAI)

---

1: **Input:** set of arms $[K]$, guarantee $\delta$
2: **Set:** $t \leftarrow 1$, $B_t \leftarrow \infty$
3: **while** $B_t > 0$ or $t \leq KT(\alpha, \delta)$ **do**
4:     **if** $\exists a \in [K]$ s.t. $N_a(t) < \max\{\sqrt{t}, T(\alpha, \delta)\}$ **then**
5:         Sample arm $A_{t+1} = \arg\min_{i \in \mathcal{U}_t} N_i(t)$ and update $\hat{\mu}_{A_t}(t)$
6:         Update confidence interval $\beta_{A_{t+1}}(t+1, \delta)$
7:     **else**
8:         $j_t \leftarrow \arg\max_{a \in [K] \setminus \hat{a}_t} \hat{\mu}_a(t) + \beta_a(t, \delta) - (\hat{\mu}_{\hat{a}_t}(t) - \beta_{\hat{a}_t}(t, \delta))$
9:         $B_t \leftarrow \max_{a \in [K] \setminus \hat{a}_t} \hat{\mu}_a(t) + \beta_a(t, \delta) - (\hat{\mu}_{\hat{a}_t}(t) - \beta_{\hat{a}_t}(t, \delta))$
10:        $A_{t+1} \leftarrow \arg\max_{\{\hat{a}_t, j_t\}} \{\beta_{\hat{a}_t}(t, \delta), \beta_{j_t}(t, \delta)\}$
11:        Pull arm $A_{t+1}$ and update means $\hat{\mu}_{A_{t+1}}$ and confidence intervals $\beta_{A_{t+1}}(t+1, \delta)$
12:     **end if**
13:     $t \leftarrow t + 1$
14: **end while**
15: Output: $\hat{a} = \arg\max_{a \in [K]} \hat{\mu}_a(t)$

---

**Estimator.** While the sample mean is widely used as an estimator for BAI algorithms, it is not robust to adversarial corruptions. It is well-known that the sample median is a more robust estimator under such circumstances. However, while the sample median is a reasonable choice for unimodal distributions or under the modified definition of the best arm, it may not be as reliable otherwise since the median and the mean could be considerably different (especially, for heavy-tailed distributions). For this purpose, our choice of the estimator strikes a balance between the sample median (to remove the outlier samples) and the sample mean (to form an estimate of the true mean). Specifically, for both the algorithms, we use the $\alpha$-trimmed mean as an estimator. This implies that given a sequence of samples $\mathbf{R}_i^t \triangleq \{R_{A_s} : s = (1, \cdots, t), A_s = i\}$ for any arm $i \in [K]$, we construct a subsequence of samples $\mathbf{Y}_i^t \triangleq \{Y_{i,j}^t\}_j$ by trimming the first and last $\alpha$-quantile of observations from

$\mathbf{R}_i^t$. Correspondingly, the $\alpha$-trimmed mean estimator is defined as the sample mean of the remaining samples, i.e.,

$$\hat{\mu}_i(t) \triangleq \frac{1}{(1-2\alpha)N_i(t)} \sum_{j=1}^{\dim(\mathbf{Y}_i^t)} Y_{i,j}^t \,, \tag{5}$$

where $\{Y_{i,j}^t\}_{j=1}^{\dim(\mathbf{Y}_i^t)}$ represent the entries of $\mathbf{Y}_i^t$, and $N_i(t)$ counts the number of times that arm $i \in [K]$ is pulled until $t$, i.e., $N_i(t) \triangleq \sum_{s=1}^{t} R_{A_s} \mathbb{1}_{\{A_s=i\}}$ .

**Gap-based algorithm for CBAI (G-CBAI).** This algorithm aims at reducing the maximum overlap between the confidence intervals of the best arm and the *most ambiguous* arm, which is defined as the arm whose confidence interval has the maximum overlap with the confidence interval of the current best arm. Similarly to the algorithm for BAI for stochastic linear MABs in [25], at each time-step, this algorithm samples the arm that maximally reduces the overlap between the current best arm and the most ambiguous arm. The sampling process stops as soon as this overlap vanishes. The design rules are formalized next.

*Arm selection rule.* There is a rich body of literature on arm selection strategies for stochastic MABs [25, 7, 26]. The problem was solved for the family of exponential bandits in [26], through a "track and stop" arm selection rule that tracks the optimal allocation of arm selections. The procedure was shown to exhibit asymptotic optimality (up to a constant factor). However, in the case of CBAI, this strategy is directly not applicable since we do not assume that the adversarial models belong to the exponential family. Hence, we propose an approach based on confidence intervals that consists of two phases: a forced exploration phase, which ensures that each arm is pulled sufficiently often such that we do not get stuck in an incorrect maximum likelihood (ML) decision of the best arm, and an exploitation phase that pulls the current best and most ambiguous arms in order to reduce the overlap in the confidence intervals between the two. At time $t$, we denote the decision about the current best arm by $\hat{a}_t$ and the most ambiguous arm by $j_t$, i.e.,

$$j_t \triangleq \arg\min_{a \in [K] \setminus \hat{a}_t} \hat{\mu}_a(t) + \beta_a(t, \delta) - (\hat{\mu}_{\hat{a}_t}(t) - \beta_{\hat{a}_t}(t, \delta)) \,, \tag{6}$$

where $\beta_i(t, \delta)$ represents the width of the confidence interval for each arm $i \in [K]$ . The choice of $\beta_i(t, \delta)$ is guided by the performance guarantee that the algorithm needs to ensure, and it is characterized in Theorem 4.2. The overlap in confidence interval is denoted by $B_t$, defined as

$$B_t \triangleq \hat{\mu}_{j_t}(t) + \beta_{j_t}(t, \delta) - (\hat{\mu}_{\hat{a}_t}(t) - \beta_{\hat{a}_t}(t, \delta)) \,. \tag{7}$$

Next, we define a constant $T(\alpha, \delta)$ as follows, which is instrumental in formalizing our arm selection strategy.

$$T(\alpha, \delta) \triangleq \frac{2}{\alpha^2} \log \frac{1}{\delta} \,. \tag{8}$$

Based on these definitions, we provide the arm selection strategy for the gap-based CBAI algorithm: at any time $t$, let $\mathcal{U}_t \subseteq [K]$ be a subset of arms defined as $\mathcal{U}_t \triangleq \{i \in [K] : N_i(t) \leq \max(\sqrt{t}, T(\alpha, \delta))\}$. At time $t$, if $\mathcal{U}_t \neq \emptyset$, the sampling strategy pulls the arm $i \in \mathcal{U}_t$ that has been pulled the least number of times. Otherwise, the algorithm pulls the arm between the best and the most ambiguous arms, whichever has the maximum confidence interval. Formally, at time $(t + 1)$ the sampling rule draws the arm $A_{t+1}$ such that

$$A_{t+1} \triangleq \begin{cases} \arg\min_{i \in \mathcal{U}_t} N_i(t) & \text{if } \mathcal{U}_t \neq \emptyset \quad \text{(forced exploration)} \\ \arg\max_{\{\hat{a}_t, j_t\}} \{\beta_{\hat{a}_t}(t, \delta), \beta_{j_t}(t, \delta)\} & \text{if } \mathcal{U}_t = \emptyset \quad \text{(exploitation)} \end{cases} \tag{9}$$

*Stopping rule.* The stopping rule is guided by the maximum overlap of the confidence intervals corresponding to the best arm and the most ambiguous arm. As soon as the two confidence intervals cease to overlap, the algorithm stops to form a decision. However, due to the concentration of the $\alpha$-trimmed mean estimator, there is a minimum number of samples that needs to be acquired before arriving at the decision. This ensures that we have enough samples to separate the uncontaminated reward samples from the outliers with a high probability. Taking into account these two aspects, the

---

**Algorithm 2** Successive elimination-based algorithm for CBAI (SE-CBAI)

1: **Set:** $t = 1$, $\mathcal{M}_t = [K]$
2: **while** $|\mathcal{M}_t| > 1$ **do**
3:     Sample arm $i \in \mathcal{M}_t$ once and produce $\hat{\mu}_i(t)$ (from all past samples)
4:     $\mathcal{M}_{t+1} \leftarrow \left\{ i \in \mathcal{M}_t : \hat{\mu}_i(t) \geq \max_{j \in \mathcal{S}_t} \hat{\mu}_j(t) - 2\gamma_t \text{ or } N_i(t) < T(\alpha, \delta) \right\}$
5:     $t \leftarrow t + 1$
6: **end while**
7: Output: $\mathcal{M}_t$

---

algorithm stops as soon as the overlap $B_t$ falls below 0, but not before each arm is pulled $T(\alpha, \delta)$ times. Formally, this is described in the following stopping rule.

$$\tau \triangleq \max \left( \frac{2K}{\alpha^2} \log \frac{1}{\delta} , \inf \left\{ t \in \mathbb{N} : B_t \leq 0 \right\} \right) . \tag{10}$$

Based on the rules specified in (5)-(10), the detailed steps of the procedure are described in Algorithm 1.

**Successive elimination-based algorithm (SE-CBAI).** In this algorithm, we maintain a set of *active* arms $\mathcal{M}_t$, which contains the arms which are not eliminated yet. At each round, we attempt to eliminate any possible suboptimal arm from the set of active arms $\mathcal{M}_t$, until we have only one element left in the set. The surviving arm is declared as the best arm. This algorithm has two phases: an exploration phase, in which each arm is explored $T(\alpha, \delta)$ times, and an exploitation phase based on the the successive elimination of suboptimal arms. This algorithm is similar to Algorithm 2 in [4], with the key difference that we use the trimmed mean estimator instead of using the sample median as the estimator. The detailed steps of the procedure are described in Algorithm 2.

## 4 Performance guarantees

In this section, we provide the performance guarantees for the G-CBAI and SE-CBAI algorithms. For this purpose, let us first define the problem complexity of any CBAI instance as

$$H \triangleq \sum_{i \in [K]} \left( \frac{\sqrt{2}\sigma}{\max\{\Delta_i, \Delta_{b^\star}\}} \right)^2 , \tag{11}$$

where we have defined $\Delta_i \triangleq (\mu_{a^\star} - U_{a^\star}) - (\mu_i + U_i)$ for every arm $i \in [K]$. The notion of problem complexity for CBAI is a generalization of the problem complexity defined for stochastic MABs, which is subsumed by setting the uncertainty terms $U_i = 0$ for every suboptimality gap $\Delta_i$ for each arm $i \in [K]$. We will observe that for the case of CBAI, the uncertainty terms $\{U_i : i \in [K]\}$ depend on the probability of attack $\varepsilon$ such that an increase in $\varepsilon$ results in a higher uncertainties, which in turn weakens the performance guarantee. First, we provide an asymptotic lower bound on the average sample complexity of any algorithm that is $\delta$-PAC in the PIBAI setting.

**Theorem 4.1** (Lower bound)**.** *In the PIBAI setting, the average sample complexity of any $\delta$-PAC procedure is asymptotically lower bounded as*

$$\lim_{\delta \to 0} \frac{\mathbb{E}[\tau]}{\log(1/\delta)} \geq H . \tag{12}$$

*Proof.* See Appendix A.  ∎

In the case of CBAI, the uncertainty term $U_i$ depends on the level of contamination $\varepsilon$ and the variance $\sigma$ of the probability density functions. Specifically, the uncertainty involved in estimating each of the arms $U_i = \Omega\left( \sigma \varepsilon \sqrt{\log \frac{1}{\varepsilon}} \right)$ for every $i \in [K]$ [23]. Using this along with Theorem 4.1, we obtain the following corollary.

**Corollary 4.1.** *In the CBAI setting, the average sample complexity of any $\delta$-PAC procedure is asymptotically lower-bounded as*

$$\lim_{\delta \to 0} \frac{\mathbb{E}[\tau]}{\log(1/\delta)} \geq \sum_{i \in [K]} \left( \frac{\sqrt{2}\sigma}{\max\{\tilde{\Delta}_i, \tilde{\Delta}_{b^\star}\} - \Omega(\sigma\varepsilon\sqrt{\log(1/\varepsilon)})} \right)^2, \tag{13}$$

*where we have defined*

$$\tilde{\Delta}_i \triangleq \mu_{a^\star} - \mu_i, \quad \text{for all } i \in [K]. \tag{14}$$

The lower-bound in (13) is a generalization of the lower bound provided in [26] for BAI in the attack-free setting. Specifically, setting $\varepsilon = 0$ in Corollary 4.1 simplifies (13) to the lower bound for BAI in the non-contaminated setup as provided in [26].

Next, we provide a non-asymptotic concentration bound for the $\alpha$-trimmed mean estimator, which is the key lemma used for analyzing the procedures proposed in Section 3. For the following concentration on the $\alpha$-trimmed mean estimator, we are considering a single arm with mean $\mu$ that we are trying to estimate. The true measure of the arm is denoted by $\mathbb{P}_\mathsf{T}$.

**Lemma 4.1** (Estimator concentration). *In the presence of an oblivious adversary, for the $\alpha$-trimmed mean estimator with $\alpha = \varepsilon/2$, there exists $T(\alpha, \delta) \in \mathbb{N}$ such that for all $t > T(\alpha, \delta)$, we have*

$$\mathbb{P}_\mathsf{T}\left\{ \left| \hat{\mu}_t - \mu \right| \geq \mathcal{O}\left( \sigma\varepsilon\sqrt{\log\frac{1}{\varepsilon}} \right) + \frac{\sigma}{1-\varepsilon}\sqrt{\frac{2}{t}\log\frac{2}{\delta}} \right\} \leq \delta. \tag{15}$$

*Proof.* See Appendix B. ∎

From the above concentration on the $\alpha$-trimmed mean estimator, we see that the bound is valid when the number of samples is at least $T(\alpha, \delta)$. This minimum number of samples ensures that the $\alpha$-trimmed mean estimator successfully distills the outliers with a high probability. In other words, given that the total number of samples is at least $T(\alpha, \delta)$, the probability that any sample from the sequence $\mathbf{Y}_i^t$ is not adversarial is no less than $1 - \delta$. Thus, with a high probability, we compute the sample mean of the uncontaminated samples, which are generated by the true model $\mathbb{P}_\mathsf{T}$. It is noteworthy that the uncertainty induced as a result of using the $\alpha$-trimmed mean estimator, i.e., $U_i = \mathcal{O}(\sigma\varepsilon\sqrt{\log(1/\varepsilon)})$ matches the universal lower bound on the uncertainty that can be induced by any estimator in the Huber's contamination model (2) under the $\sigma$-sub-Gaussian assumption [23]. We remark that the uncertainty term can be improved by using the empirical median as an estimator, in which case it is of the order $\mathcal{O}(\varepsilon/(1-\varepsilon))$ [4]. This is, albeit, viable at the expense of stricter assumptions on the class of bandit instances. Lemma 4.1 is a significant improvement over the existing concentrations on the trimmed mean estimator. An existing bound that is most relevant to the scope of this paper can be found in [[19], Theorem 1]. Specifically, the concentration bound involves a $\log t$ term, which increases significantly for large $t$. The tightness in the bound provided in Lemma 4.1 is a result of considering the uncertainty $U_i$ in estimating the true mean. This is supported by proving that the trimmed mean achieves the lower bound on the minimum uncertainty in the sub-Gaussian setting.

The next theorem characterizes an appropriate choice of the width of the confidence intervals $\beta_i(t, \delta)$, such that a $\delta$-PAC guarantee can be ensured by Algorithm 1 in the PIBAI setting.

**Theorem 4.2** ($\delta$-PAC). *For any $\delta \in (0, 1)$, the procedure proposed in Algorithm 1 is $\delta$-PAC in the PIBAI setting for the choice of exploration threshold*

$$\beta_i(t, \delta) \triangleq \frac{\sigma}{1-\varepsilon}\sqrt{\frac{2}{N_i(t)}\log\frac{(K-1)Ct^\beta}{\delta}}, \quad \text{for all } i \in [K], \tag{16}$$

*for any $\beta > 1$, where $C \triangleq (1 + (1-\beta)^{-1})$.*

*Proof.* See Appendix C. ∎

Next, we provide an upper bound on the average sample complexity of Algorithm 1 in the asymptote of diminishing error probability $\delta \to 0$.

**Theorem 4.3** (Sample complexity). *In the asymptote of $\delta \to 0$ and for any $0 < \varepsilon < 1/2$, the average sample complexity of the procedure proposed in Algorithm 1 is upper bounded as*

$$\lim_{\delta \to 0} \frac{\mathbb{E}[\tau]}{\log(1/\delta)} \leq \max\left\{\frac{8K}{\varepsilon^2}, \, 64\beta H\right\}. \tag{17}$$

*Proof.* The details of the proof are provided in Appendix D. The crux of the analysis lies in the fact that the additivity property of suboptimality gaps is not satisfied in the CBAI setting [[4], Remark 4], which becomes a key component of the analysis for the attack-free counterpart of Algorithm 1. Specifically, it can be shown that in the attack-free setting, a fixed exploration phase of one sample per arm followed by the exploitation phase is sufficient to prove that Algorithm 1 is instance-optimal up to logarithmic factors. The general analysis follows a similar line of argument presented in Appendix D, combined with the fact that in the attack-free setting ($U_i = 0$ for every $i \in [K]$), $\mu_i - \mu_j = \Delta_j - \Delta_i$. However, in the CBAI setting, this relationship is no longer valid. The exploration phase of Algorithm 1 circumvents this difficulty in the analysis, by noting that visiting each arm at least $\sqrt{t}$ number of times ensures that we have $\Delta_{\hat{a}_t} = \Delta_{a^*}$ asymptotically as $\delta \to 0$. This is the key ingredient that differentiates the analysis of Algorithm 1 in the CBAI context compared to the attack-free counterpart. This is also the key reason that we are only able to provide asymptotic results on the average sample complexity for the CBAI setting, as opposed to non-asymptotic high probability upper bounds on the sample complexity matching up to logarithmic factors in the attack-free scenario. ∎

Theorem 4.1 and Theorem 4.3 establishes that Algorithm 1 is asymptotically optimal (up to constant factors) in the limit of $\delta \to 0$ for any bandit instance satisfying the property that $H > K/8\beta\varepsilon^2$. We remark that the first term in the sample complexity bound in (17) is attributed to the property of the estimator, and hence, is a penalty that we need to pay as a result of the contamination. Note that in the attack-free scenario ($\varepsilon = 0$), the trimmed mean estimator simplifies to the sample-mean estimator, in which case it is well-known that we do not have the first term inside the maximization. Hence, the above upper-bound is only valid for $\varepsilon > 0$. The next corollary explicates the relationship between the average sample complexity and the fraction of adversarial samples $\varepsilon$. This is a direct result of Lemma 4.1 and Theorem 4.2.

**Corollary 4.2.** *In the asymptote of $\delta \to 0$, there exists a constant $C_1 \in \mathbb{R}_+$ such that the average sample complexity of Algorithm 1 is upper bounded as*

$$\lim_{\delta \to 0} \frac{\mathbb{E}[\tau]}{\log(1/\delta)} \leq \max\left\{\frac{8K}{\varepsilon^2}, \, 64\beta \sum_{i \in [K]} \left(\frac{\sqrt{2}\sigma}{\max\{\tilde{\Delta}_i, \tilde{\Delta}_{b^*}\} - 2C_1\sigma\varepsilon\sqrt{\log(1/\varepsilon)}}\right)^2\right\}, \tag{18}$$

*where we have defined*

$$\tilde{\Delta}_i \triangleq \mu_{a^*} - \mu_i \quad \text{for all } i \in [K]. \tag{19}$$

Next, we provide the performance guarantees corresponding to Algorithm 2. First, we provide a choice of $\gamma_t$ that ensures that Algorithm 2 is $\delta$-PAC in the PIBAI setting.

**Theorem 4.4** ($\delta$-PAC). *For any $\delta \in (0, 1)$, Algorithm 2 is $\delta$-PAC in the PIBAI setting for the following choice of the exploration threshold.*

$$\gamma_t \triangleq \frac{\sigma}{(1-\varepsilon)}\sqrt{\frac{2}{t}\log\frac{Kt^2\pi^2}{12\delta}}. \tag{20}$$

*Proof.* See Appendix E. ∎

Finally, we provide a probabilistic upper bound on the sample complexity of Algorithm 2.

**Theorem 4.5** (Sample complexity). *With a probability not smaller than $1-\delta$ and for any $0 < \varepsilon < 1/2$, the sample complexity of Algorithm 2 is upper bounded as*

$$\tau \leq \max\left\{\frac{8K}{\varepsilon^2}\log\frac{1}{\delta}, \, \mathcal{O}\left(\sum_{i \in [K] \setminus a^*} \frac{1}{\Delta_i^2}\log\frac{K}{\delta\Delta_i}\right)\right\}. \tag{21}$$

*Proof.* See Appendix F. ∎

Theorem 4.5 shows that the sample complexity of Algorithm 2 is instance optimal up to logarithmic factors. This follows from the fact that $\sum_{i \in [K] \setminus a^\star} 1/\Delta_i^2 < H$.

## 5 Experiments

In this section, we evaluate the empirical performance of the G-CBAI and SE-CBAI algorithms and compare it to that of the existing algorithms. We provide synthetic and real-world experiments that consider Gaussian bandits as a common framework for comparison with the median-based successive elimination framework in [4], although our algorithms work for any general sub-Gaussian bandit environment too.

**Experiments using synthetic data.** We consider a Gaussian bandit instance with $K = 4$, and the true mean vector is $\boldsymbol{\mu} \triangleq [2.5,\ 2.3,\ 2,\ 0.6]$. For comparison, we test four strategies: (i) gap-based algorithm (Algorithm 1), (ii) successive elimination-based algorithm (Algorithm 2), (iii) Algorithm 1, along with a random sampling strategy that selects any arm $a \in [K]$ with a uniform probability, and (iv) the successive elimination-based algorithm in [4], which uses the empirical median as an estimator. Figure 1 depicts the average sample complexity versus varying confidence levels $\delta$, when the attack probability $\varepsilon$ is set to $\varepsilon = 0.1$. We observe that in this experiment, the median-based successive elimination strategy [4] has a slightly better performance compared to the proposed trimmed mean estimator with Algorithm 2. This improvement is a consequence of the fact that we are using a Gaussian bandit for which the empirical median yields a better error guarantee (of the order $\mathcal{O}(\frac{\varepsilon}{1-\varepsilon})$) compared to the trimmed mean ($\mathcal{O}(\varepsilon\sqrt{\log(1/\varepsilon)})$). However, our proposed algorithms work for any sub-Gaussian bandit instance, which is not the case for the median-based strategy. Figure 2 shows how the sample complexity scales with increasing levels of contamination $\varepsilon$.

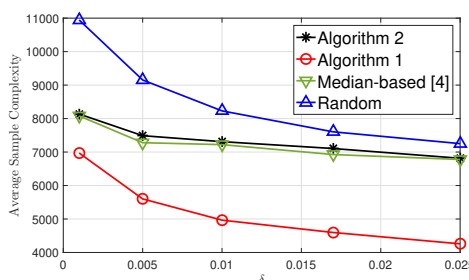
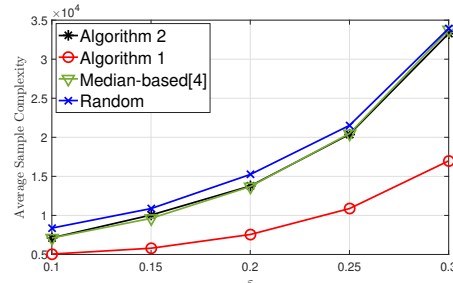

| Figure 1: Synthetic data: $\mathbb{E}[\tau]$ versus $\delta$. | Figure 2: Synthetic data: $\mathbb{E}[\tau]$ versus $\varepsilon$. |

Algorithms compared: (i) gap-based algorithm (Algorithm 1), (ii) successive elimination-based algorithm (Algorithm 2), (iii) median-based successive elimination proposed in [4], and (iv) random arm selection with the stopping rule of Algorithm 1

Our observation indicates that even though not supported by theory, for the proposed gap-based strategy, a confidence interval of the form $\beta_a(t, \delta) = \sigma\sqrt{2/N_a(t)\log(\log t/\delta)}$ for every arm $a \in [K]$ is over-conservative and meets the prescribed guarantee on decision confidence. The looseness in the theoretical confidence interval has also been observed in the algorithm in [26] for BAI in stochastic multi-armed bandits, where the empirical performances are compared using a tighter exploration threshold (of the order of $\log((\log t)/\delta)$).

**Experiments using real data.** We use two real-world datasets, namely the New Yorker Caption Contest (NYCC) dataset and the PKIS2 dataset for comparing our algorithms to the existing ones. The NYCC dataset is a collection of cartoons and captions fitting the cartoons, along with user ratings as to "how funny" the cartoons (along with the corresponding captions) are. In our experiment, we aim to find the funniest cartoon among a given subset of them, while observing noisy user ratings for each of them. On the other hand, the PKIS2 dataset is a collection of protein kinase, and several kinase inhibitors. The aim of this experiment is to find the most effective inhibitor against a targeted kinase. Other details about these datasets and the experiments are presented in Appendix G. For both

the datasets, we plot the sample complexity versus varying levels of decision confidence $\delta$ when the probability of attack is set to $\varepsilon = 0.1$. These results are depicted in Figure 3a and Figure 3b .

Our experiments show that the G-CBAI algorithm outperforms all existing algorithms for both the real-world and synthetic datasets owing to the tightness in the confidence intervals (and hence, the stopping criterion). Furthermore, it is noteworthy that as we increase $\delta$, the gap between the empirical performance of the random selection strategy and the successive elimination based strategy reduces. This is due to the fact that the stopping criterion for the randomized selection procedure is the same as that of the gap-based algorithm, and the improvement in performance of the randomized procedure is due to the tightness of the confidence intervals used in Algorithm 1. We also note that using the confidence interval defined in (16) for the G-CBAI algorithm, the SE-CBAI algorithm outperforms the gap-based method. More details are provided in Appendix G. Specifically, we provide more experiments comparing the trimmed mean estimator to other estimators (see Figures 5a, 5b and 5c) in order to establish the superior performance of the trimmed mean estimator in various settings.

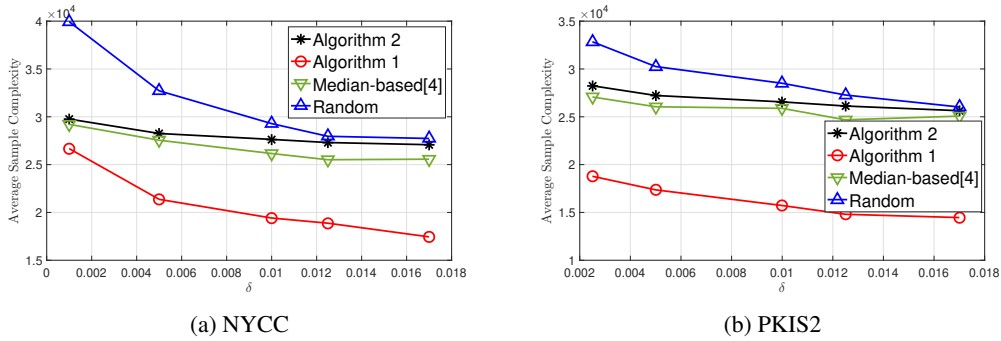

(a) NYCC                                    (b) PKIS2

Figure 3: Experiments with **real data**: $\mathbb{E}[\tau]$ versus $\delta$ plotted for (i) G-CBAI algorithm (Algorithm 1), (ii) SE-CBAI algorithm (Algorithm 2), (iii) median-based successive elimination proposed in [4], and (iv) random arm selection with stopping rule of Algorithm 1.

## 6    Conclusions

In this paper, we have investigated the problem of best arm identification for stochastic multi-armed bandits under adversarial corruptions. Specifically, we have assumed that with a probability $\varepsilon$, each reward sample is replaced by a sample drawn from an adversarial distribution. Under this framework, we have proposed two algorithms for best arm identification and have analyzed their optimality properties in terms of the average sample complexity. Specifically, a gap-based algorithm has been shown to be asymptotically optimal up to constant factors, while a successive elimination-based algorithm is shown to be optimal up to logarithmic factors. Finally, we have performed experiments with synthetic as well as real-world data to compare the empirical performance of the proposed algorithms to existing ones. The experiments have shown the superior empirical performance of the G-CBAI algorithm in both synthetic as well as real world settings, while the SE-CBAI algorithm is seen to exhibit superior performance with the confidence intervals supported by theory. Our experiments have also shown the advantage of using the trimmed mean estimator over other popular estimators in different bandit environments. One limitation of this investigation is that we have considered an oblivious adversary, while in practice, adversarial models could be more powerful. Analyzing contaminated best arm identification for stronger adversaries (e.g., prescient and malicious adversaries) for the class of sub-Gaussian bandits is open for further investigation.

## Acknowledgments and Disclosure of Funding

This work was supported by the Rensselaer-IBM AI Research Collaboration (http://airc.rpi.edu), part of the IBM AI Horizons Network (http://ibm.biz/AIHorizons)

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
