# OpenReview forum: "Mean-based Best Arm Identification in Stochastic Bandits under Reward Contamination"
_NeurIPS.cc/2021/Conference — NeurIPS 2021 Poster_

### Official Review · Reviewer_82x7 · 2021-07-13

**Rating:** 7
**Confidence:** 4

**Summary:**

This work studies the BAI problem with the fixed-confidence setting in the contaminated stochastic bandits, where the rewards can be corrupted with probability $\varepsilon$. It proposes two algorithms and derives upper bounds on their sample complexity. It also derives a lower bound to show the tightness of the bounds. Moreover, it shows numerical results in the end.


**Limitations And Societal Impact:**

The papers is generally well written. However, it would be better if the notations can be introduced before they are presented. I feel hard to find the definitions. The major suggestions are about the following:

1. Algorithm 1:    $\mathcal{U}_t$, $\hat{a}_t$, $\beta_a(t,\delta)$ are defined a bit far away from the algorithm.
2. Where is the the definition of $U_a$?


**Main Review:**

Generally speaking, this work is well-organized and it is interesting in the sense the mean-based BAI problem with fixed-confidence setting has not been explored for the contaminated bandits before.

Besides, it shows that the theoretical and numerical performance of the two proposed algorithms. However, the analytical methods seem to be quite standard. It would be better if the author(s) can point out the challenge to derive the bounds.

Lastly, it would be better if the notations can be introduced before they are presented.


=========

Thanks for the detailed response from the author(s). I have read them and want to increase the rating from 6 to 7.

**Time Spent Reviewing:**

1

---

> ### Author Response · Authors · 2021-08-10
> **Response to reviewer 82x7 of the Paper9665**
>
> We thank the reviewer for the review. Please find our response to the specific questions below:
>
> **Comment**: It would be better if the author(s) can point out the challenge to derive the bounds.
>
> **Response**: The principle challenge in deriving the bounds (confidence bounds for estimator concentration and sample complexity bounds for the proposed algorithms) comes from the contamination model, which renders it impossible to estimate the mean reward of each arm any better than an uncertainty interval $U_i$ around the true mean, even using an infinite number of samples from that arm. This leads to the following issues in the analysis:
>
> i) *Estimator concentration*: While the concentration result relies on standard inequalities for sub-Gaussian distributions, the corrupted samples inflict a deviation $U_i$ from the true mean for each arm $i\in[K]$. This motivates the use of tools from the robust statistics literature (specifically, the resilience property of sub-Gaussian distributions) to upper bound the maximum deviation that the trimmed-mean estimator suffers due to the contamination. We observe that this deviation matches the lower bound on the error-guarantee on sub-Gaussian distributions [28]. To further reinforce the gain obtained by using the trimmed-mean estimator (instead of the sample mean, as used in optimal algorithms for BAI), we have performed more experiments with the algorithms for BAI operating on contaminated samples. Specifically, we have simulated the successive elimination algorithm and the gap-based algorithm (attack-free BAI counterparts) and compared them against the proposed Algorithm 2 and Algorithm 1 with a corruption level of $\varepsilon=0.1$. The results are as follows:
>
>
> | $\delta$               | 0.025         | 0.017         | 0.01          | 0.005         | 0.001         |
> |------------------------|---------------|---------------|---------------|---------------|---------------|
> | Successive Elimination | 6988.8        | 6994.5        | 7359.4        | 7491.7        | 8002.9        |
> | Algorithm 2            | **6943.4** | **6992.4** | **7128.5** | **7429.9** | **7966.9** |
>
> ---
>
> | $\delta$                | 0.025      | 0.017    | 0.01       | 0.005      | 0.001      |
> |-------------------------|------------|----------|------------|------------|------------|
> | Gap-based (attack-free) | 4328.4     | 4529.5   | 5059.5     | 5620.3     | 6959.1     |
> | Algorithm 1             | **4243.5** | **4521** | **4995.1** | **5572.8** | **6900.2** |
>
> ---
>
> The above results clearly demonstrate the advantage that the trimmed-mean estimator provides in the contaminated setting over the sample-mean.
>
> ii) *Sample complexity upper bound for Algorithm 1 (Theorem 4.3)*:  The analysis for the attack-free counterpart of Algorithm 1 heavily relies on the property of the additivity of suboptimality gaps [24]. Specifically, the additivity of the suboptimality gap implies that $\Delta(i,j) = \Delta_i - \Delta_j$, where we have defined $\Delta(i,j) \triangleq \mu_j - \mu_i$ and $\Delta_i \triangleq \mu_{a^\star} - \mu_i$ for any arm $i,j\in[K]$. However, it can be readily verified that this property fails to hold in the contaminated setup due to the uncertainty intervals $U_i$. To get around this difficulty, we resort to a forced exploration phase of at least $\sqrt{t}$ pulls per arm in order to ensure that we have sufficient samples to estimate the means of each arm (and hence, the best arm) within the uncertainty interval up to a sufficient level of tolerance. This ensures that $\Delta_{\hat a_t}\rightarrow\Delta_{a^\star}$ asymptotically. Such fixed exploration phases are not required for gap-based  algorithms in the attack-free setting, but it becomes a key factor in obtaining the sample complexity bound in the CBAI setting. We refer the reviewer to the proof of Theorem 4.3 and Appendix C for the details.
>
> **Comment**: it would be better if the notations can be introduced before they are presented.
>
> **Response**: Thanks for the suggestions. In the revised manuscript we will ensure that all notations are defined upfront and where they are used to enhance readability.

---

> > ### Author Response · Authors · 2021-08-31
> > **Kindly inquiring if you had a chance to examine the response**
> >
> > We'd like to again thank your for your evaluation. This is a kindly follow-up to inquire whether your concerns have been addressed. We'd be grateful if you re-evaluate your rating of the paper, if you believe we've addressed your concerns.

---

> > > ### Comment · Reviewer_82x7 · 2021-09-01
> > > **Increase the rating to 7**
> > >
> > > Thanks for your response. I tend to increase the rating from 6 to 7. Good luck!

---

### Official Review · Reviewer_xUqy · 2021-07-14

**Rating:** 6
**Confidence:** 4

**Summary:**

The authors consider the problem of best arm identification in contaminated stochastic bandits. More precisely, in this framework, the reward probability distribution of each arm may be arbitrarily corrupted with probability $\varepsilon$. The authors focus on the fixed budget setting and propose two different algorithms with theoretical guarantees. Both algorithms use an $\alpha$-trimmed mean estimator to tackle the corruption problem.  The first algorithm (see Algorithm 1) is a gap-based algorithm that relies on the overlap between the confidence intervals of the best empirical arms and the remaining arms in its sampling, stopping, and decision rules. The second algorithm is an elimination-based algorithm, that eliminates arms as soon as they are deemed suboptimal. The authors complement their theoretical findings with experimental results.

**Ethical Concerns:**

I cannot think of any ethical issues with this paper.

**Limitations And Societal Impact:**

I believe the authors have addressed these points adequately.  Perhaps there could be potentially negative impacts of the bandit framework in recommender systems. I invite the authors to look at the so-called bubble effect.

**Main Review:**


The problem is interesting and the authors have successfully adapted existing algorithmic ideas in the bandit literature such as successive elimination and gap-based ideas with confidence intervals. It seems that the main challenge is to handle the corrupted samples when the algorithms build their estimates. The authors propose the $\alpha$-trimmed estimator with a theoretical guarantee. However, I am not fully convinced why this is better or worse than the median estimator used by Altschuler et al. in [4] (the authors cite this paper). The authors argue that using the median estimator might no be a good idea if the distributions are heavy-tailed or not unimodal, but after all in the considered setting the distributions are Gaussian. Besides, in the presented experiments the performance of algorithm 2 and that of Altshuler et al. [4] that uses the median estimator performs similarly. Perhaps the author should consider an experiment with synthetic data where the median estimator truly fails. Could the authors elaborate further on the matter?

The authors provide a lower bound (see Theorem 4) which is very much appreciated. However, it is a bit disappointing that the provided lower bound is essentially the same as the one provided for the classical bandit setting by Garivier et al [26], and does not provide any quantitative insight on how difficult is the contaminated setting in comparison with the classical setting in terms of sample complexity. After all, the obtained guarantees do depend on the contamination level $\varepsilon$, and would hope to obtain a lower bound that depends on $\varepsilon$ as well.


The authors provide theoretical guarantees on the sample complexity for both algorithms. However, the expressed guarantee for algorithm 1 (gap-based algorithm) is in the expected number of samples and is asymptotic in $\delta$ (see Theorem 4.3), while the expressed guarantee for algorithm 2 (elimination based algorithm) is in high probability and is non-asymptotic in $\delta$. The fact that these two guarantees are different blurs the comparison between the two algorithms. Could the authors explain why they cannot obtain similar types of guarantees? are there any challenges in obtaining similar guarantees? the experiments also indicate that algorithm 1 is superior compared to algorithm 2 (see Figure 1, 2, and 3), although some experiments indicate the opposite (see Figure 4).

In algorithm 1, the authors use the thresholds $\beta_a(t,\delta) = \sigma\sqrt{ 2/N_a(t)\log(\log(t/\delta))}$ which is not theoretically proven. Could this be the reason why algorithm 1 outperforms the other algorithms? If this is the case, then maybe the comparison is unfair. Have the authors checked the performance of algorithm 1 with the theoretically proven threshold? For more refined concentration bounds, perhaps the author should look at the paper titled "Mixture Martingales Revisited with Applications to Sequential Tests and Confidence Intervals" by Kaufmann and Koolen. Furthermore, it makes more sense to compare in the experiments the performance of optimal algorithms for best arm identification without corruption rather than with random arm selection.


**Time Spent Reviewing:**

10 hours

---

> ### Author Response · Authors · 2021-08-10
> **Response to reviewer xUqy of the Paper9665**
>
> We thank the reviewer for the detailed review and thoughtful comments.
>
> **Comment**: I am not fully convinced why this is better or worse than the median estimator used by Altschuler et al. in [4] (the authors cite this paper).
>
> **Response**: We establish the suitability of the trimmed-mean estimator over median estimators by the following three points:
>
> 1- Our paper is the first to provide an algorithm with provable guarantees for the BAI problem in the $\sigma$-sub-Gaussian setup with data contamination. As all the reviewers have pointed out, this is an important setting and merits independent investigation.
>
> 2- We acknowledge the reviewer’s comment that the relative performance of median versus trimmed-mean estimator is unclear. Even though there is no theoretical proof that median-based estimators cannot be optimal in all settings, there is strong evidence that they are not optimal in a wide range of models. The reason is the following: by the law of large numbers, the empirical median estimator converges to the true median asymptotically. Thus, if the median and the mean for the distributions are very different (for example, in the case of heavy-tailed distributions), the empirical median estimate will have a large deviation from the mean, even if we have an infinite number of samples.
>
> 3- Even for the settings that a median-based estimator can be optimal, we do not believe that should be considered a weakness of the trimmed-mean estimators. It is almost impossible to prove that an algorithm is uniquely optimal, and our bottom line observation is that for a problem hitherto unsolved, we are providing an algorithm with provable guarantees, but we also recognize that there can be other alternatives, including median-based ones for special settings.
>
> **Comment**: after all in the considered setting the distributions are Gaussian.
>
> **Response**: We would like to clarify that we are considering a much broader class of distributions (sub-Gaussian) not Gaussian.  That is one setting in which the median and mean can be considerably distinct, and median-based estimators cannot be optimal. Our setup subsumes popular special cases such as Bernoulli bandits and exponential bandits for which the mean and the median are not the same., and hence the distributions may not be unimodal in general.
>
> **Comment**: ​​ in the presented experiments the performance of algorithm 2 and that of [4] that uses the median estimator performs similarly.
>
> **Response**: We, respectively, disagree. We would like to refer the reviewer to the Figure 4(b) in Appendix G, which uses the exponential bandit setup and it is observed that Algorithm 2 outperforms the median-based strategy used in [4]. Besides that our Algorithm 1 significantly outperforms the approach of [4].
>
> **Comment**: authors provide a lower bound (see Theorem 4) which is very much appreciated. However, it is a bit disappointing that the provided lower bound is essentially the same as the one provided for the classical bandit setting by [26] …
>
> **Response**: Our lower bound is different from that in Garivier et al. [26], despite some similarities. More specifically, by setting $U_i=0$ our lower bound reduces to and subsumes that of [26].  But our bound is more general.
>
> **Comment**: After all, the obtained guarantees do depend on the contamination level $\varepsilon$ and would hope to obtain a lower bound that depends on $\varepsilon$  as well.
>
> **Response**: Our lower bound indeed depends on $\varepsilon$. Our lower bound is characterized by $H$, which as defined in (11) explicitly depends on $\varepsilon$ through the terms $U_i=\Omega\bigg(\sigma\varepsilon\sqrt{\log\frac{1}{\varepsilon}}\bigg)$ [28].
>
> **Comment**: Does not provide any quantitative insight on how difficult is the contaminated setting in comparison with the classical setting in terms of sample complexity.
>
> **Response**: Thanks for the suggestion. There are indeed important  technical differences. In the revised paper we will include those. In summary, the key difference in the analysis of the lower bound compared to the analysis in [26] comes from the fact that the true mean of any arm cannot be estimated in the contaminated setting even if we have an infinite number of samples from that arm [4]. Rather, each arm mean can be estimated up to a level of uncertainty $U_i$ (which is a function of $\varepsilon$). The analysis has two main steps:
>
> i) First, we find a BAI instance such that the CBAI instance requires at least as many samples to ensure the $\delta$-PAC guarantee as the BAI instance for any algorithm. Such a BAI instance is characterized by the parameters $\mu_i$ and $U_i$ for each arm $i\in[K]$ of the CBAI instance. This is the key difference in lower bounding the CBAI instance.
>
> ii) Next, we use the result from [26] in order to lower bound the problem complexity of the BAI instance.
>
> Substituting the values of $U_i$ in the problem complexity H, we can obtain a corollary to Theorem 4.1 in the same fashion as that of Corollary 4.1, which depends on the level of contamination $\varepsilon$. Specifically, we have that $$\lim\limits_{\delta\rightarrow 0}\frac{\mathbb{E}[\tau]}{\log(1/\delta)}\geq \sum\limits_{i\in[K]}\frac{2\sigma^2}{(\max\({\tilde\Delta_i,\tilde\Delta_{b^\star}\})-\Omega(\sigma\varepsilon\sqrt{\log(1/\varepsilon)})^2}$$, where we have defined $\tilde\Delta_a\triangleq\mu_{a^\star}-\mu_a$.
>
> **Comment**: Could the authors explain why they cannot obtain similar types of guarantees? are there any challenges in obtaining similar guarantees?
>
> **Response**:  Thanks for the thoughtful suggestion. As the reviewer correctly suggested, there are technical challenges due to which finding a unifying theme for theoretical comparisons even in the simpler settings (e.g., non-contaminated models) is difficult (and non-existent in the current literature). In this paper, for completeness, we had provided two algorithms with different natures (in line with the general approaches to the BAI algorithms). The difference in the sample complexity expressions (average sample complexity versus high probability bound) is owed to the difficulty of analyzing Algorithm 1 (line 268). Specifically, this is due to the fact that the additivity property of suboptimality gaps is not satisfied in the contaminated setup, which is a key component for deriving the high probability sample complexity upper bound for the attack-free counterpart of the gap-based algorithm [24]. Specifically, the additivity of the suboptimality gap implies that $\Delta(i,j) = \Delta_i - \Delta_j$, where we have defined $\Delta(i,j) \triangleq \mu_j - \mu_i$ and $\Delta_i \triangleq \mu_{a^\star} - \mu_i$ for any arm $i,j\in[K]$. It can be readily verified that this property fails in the contaminated setup due to the uncertainty intervals $U_i$. To get around this difficulty, we resort to a forced exploration phase of at least $\sqrt{t}$ pulls per arm in order to ensure that we have sufficient samples to estimate the means of each arm (and hence, the best arm) within the uncertainty interval up to a sufficient level of tolerance. This ensures that $\Delta_{\hat a_t}\rightarrow\Delta_{a^\star}$ asymptotically. Such fixed exploration phases are not required for gap-based  algorithms in the attack-free setting, but it becomes a key factor in obtaining the sample complexity bound in the CBAI setting. We refer the reviewer to the proof of Theorem 4.3 and Appendix C for the details.
>
>
> **Comment**: In algorithm 1, the authors use the thresholds  …   which is not theoretically proven. Could this be the reason why algorithm 1 outperforms the other algorithms? If this is the case, then maybe the comparison is unfair.
>
> **Response**:
>
> 1- We acknowledge that the thresholds used for the two algorithms are different.  Indeed, we are not arguing that the performance of each might or might not change with other choices of the thresholds. However, what we analytically prove depends on these thresholds. Indeed, the choice of the confidence interval is the key reason why Algorithm 1 outperforms Algorithm 2.
>
> 2- To ensure the fairness in comparisons, we are selecting the thresholds such that they ensure the same $\delta$-PAC guarantee for both algorithms. On the other hand, using the same thresholds does not necessarily lead to a fair comparison. Besides that, comparisons using the same thresholds for both algorithms is not even feasible, since for Algorithm 2, the threshold $\gamma_t$ cannot depend on the number of times any specific arm $a\in[K]$ is pulled ($N_a(t)$).
>
> **Comment**: Have the authors checked the performance of algorithm 1 with the theoretically proven threshold?
>
> **Response**: We have also plotted the theoretically proven confidence bounds in Appendix G (see Fig. 4). As suggested by the reviewer, we have performed further experiments with the algorithms for BAI (attack-free counterparts) operating on contaminated samples with a corruption level of $\varepsilon=0.1$. The results are as follows:
>
> | $\delta$               | 0.025         | 0.017         | 0.01          | 0.005         | 0.001         |
> |------------------------|---------------|---------------|---------------|---------------|---------------|
> | Successive Elimination | 6988.8        | 6994.5        | 7359.4        | 7491.7        | 8002.9        |
> | Algorithm 2            | **6943.4** | **6992.4** | **7128.5** | **7429.9** | **7966.9** |
>
> ---
>
> | $\delta$                | 0.025      | 0.017    | 0.01       | 0.005      | 0.001      |
> |-------------------------|------------|----------|------------|------------|------------|
> | Gap-based (attack-free) | 4328.4     | 4529.5   | 5059.5     | 5620.3     | 6959.1     |
> | Algorithm 1             | **4243.5** | **4521** | **4995.1** | **5572.8** | **6900.2** |
>
> ---
> The above results clearly demonstrate the advantage that the trimmed-mean estimator provides in the contaminated setting over the sample-mean.

---

> > ### Comment · Reviewer_xUqy · 2021-08-25
> > **Further comments**
> >
> > Thanks for your clarification and responses. I still have a few remarks and some questions which I listed below.
> >
> > **Provable guarantees for PIBAI.** Regarding your claim that you are the first to provide an algorithm with provable guarantees for the BAI problem in the $\sigma$-sub-Gaussian setup with data contamination. Didn’t Altshuler et al. [4] consider the same problem and provide an algorithm with provable guarantees (See Theorem 2 for the simple algorithm and Theorem 3 for the instance-adaptive algorithm in [4])? In what sense do you mean that you are the first?
> >
> > **$\alpha$-trimmed vs. median.** Forgive my imprecision there when I said *gaussian*, I understand you are looking at distributions that have sub-gaussian tails ($\mathbb{P}( X > t ) \lesssim e^{-\alpha t^2}$). You are claiming that the $\alpha$-trimmed estimator might be a better choice than the median estimator when dealing with heavy-tailed distributions. My concern here is that even if this is the case it is irrelevant to the setting you study (distributions with sub-gaussian tails).  Therefore, I would suggest perhaps strengthening this claim with some theoretical results or at least some experiments.
> >
> > **Lower bound and dependence on $\varepsilon$.** I think it would be useful to emphasize the dependency of the complexity $H$ on $\varepsilon$, but also on the problem instance to clarify that what you have is an instance-specific lower bound that also depends on $\varepsilon$. Thanks for clarifying this dependence, I am however a bit curious on how $U_i$ depends on the arm $i$, and whether that affects the dependence on $\epsilon$. Also, I believe that the authors in [4] also unveil that there is a bias that we cannot avoid when estimating the means (the U_i) and list this as one of their contributions.

---

> > > ### Author Response · Authors · 2021-08-26
> > > **Response to further comments by reviewer xUqy**
> > >
> > > Thanks for the detailed questions. Please find our responses below.
> > >
> > > **Comment:** Didn’t Altshuler et al. [4] consider the same problem and provide an algorithm with provable guarantees (See Theorem 2 for the simple algorithm and Theorem 3 for the instance-adaptive algorithm in [4])? In what sense do you mean that you are the first?
> > >
> > > **Response**: Thanks for the question. The short answer is no. Indeed [4] provides algorithms with provable guarantees. But those are for a problem distinct from what is generally considered as the BAI problem: To elaborate we note that we have two substantial and fundamental distinctions from [4] in the statement of the problem, scope of the problem, and the nature of the guarantees:
> > >
> > > 1. **Setup**: The BAI problems generally aim to identify the arm with the largest *mean* value.  [4], however, adopts a definition distinct from this canonical model and aims to identify the art with the largest *median*. This work is the first to consider the *standard* definition of the BAI problem under model contamination.
> > >
> > > 2. **Data model**: Another significant distinction is that we consider the *standard* sub-Gaussian model. In contrast, [4] focuses on a considerably more restricted class of models (specified in [4, Definition 2]) which does not include many popular and widely-analyzed classes of bandit instances (e.g., heavy-tailed continuous models and all discrete models such as the Bernoulli). For convenience in checking these, we are copying Definition 2 of [4]:
> > >
> > > (Definition 2, [4]): *For any $\bar{t}\in\bigg(0,\frac{1}{2}\bigg)$ and $B>0$, let $F_{\bar{t},B}$ be the family of distributions $F$ that have a unique median and satisfy
> > > $\lvert F(x_1) - F(x_2)\rvert \geq \frac{1}{Bm_2(F)}\lvert x_1 - x_2\rvert$
> > > for all $x_1,x_2\in[Q_{L,F}(\frac{1}{2}-\bar{t}),Q_{R,F}(\frac{1}{2}+\bar{t})]$.*
> > >
> > > In the above definition, $m_2(F)$ is the median absolute deviation. The definition essentially refers to the class of cumulative distribution functions (CDFs) that are *not too flat* near the median, and increase at least linearly. We are the first to study the contaminated best arm identification (CBAI) problem in the **sub-Gaussian** setup.
> > >
> > > **Comment:** You are claiming that the $\alpha$-trimmed estimator might be a better choice than the median estimator when dealing with heavy-tailed distributions...Therefore, I would suggest perhaps strengthening this claim with some theoretical results or at least some experiments.
> > >
> > > **Response**:  Thanks for the suggestion. We have performed more experiments as suggested by the reviewer, which we will discuss. However, before that, let us clarify two points:
> > >
> > > 1. There is no known result in the existing literature for the *standard* BAI problem (i.e., best mean) in the *sub-Gaussian* setup under model contamination. Our paper is the first to establish some results for these canonical settings. Again, we would like to emphasize that the existing work in [4] focuses on a *non-standard* modified BAI problem under a very limited data model with strict regulatory conditions on the CDF.
> > >
> > > 2. Having $\alpha$-trimmed- versus median-based estimators is not a main point of this paper. We provide a discussion on comparing them to shed light on our rationale for choosing the $\alpha$-trimmed approach. However, we do not intend (and believe it is even impossible) to rule out that other approaches (including median-based) might enjoy some form of optimality.
> > >
> > > As suggested by the reviewer, we have performed more experiments to compare the performance of the trimmed mean vs. the median as an estimator, both applied to Algorithm 2 that we have proposed.
> > >
> > > Experiment 1: The advantage of trimmed mean-based estimators: We first consider a simple bandit instance with K=2 arms, where the arms generate rewards drawn from a log-normal distribution (which is a heavy-tailed distribution). The parameters used for the two arms are $\mu = [1,1.05]$ and $\sigma = [1,1.2]$. Clearly, in this case, the trimmed mean outperforms the median as an estimator. This empirically confirms our rationale that trimmed-mean is a better choice compared to median for heavy-tailed distributions. Both the above experiments use a contamination level of $\varepsilon=0.1$.
> > >
> > > | $\delta$                  | 0.025       | 0.017       | 0.01        | 0.005       | 0.001       |
> > > |---------------------------|-------------|-------------|-------------|-------------|-------------|
> > > | Algorithm 2: trimmed mean | **123.784** | **126.636** | **133.896** | **168.496** | **186.272** |
> > > | Algorithm 2: median       | 839.2       | 982.8       | 1024.3      | 1496.8      | 1604.6      |
> > >
> > > Experiment 2: The advantage of median-based estimators: To emphasize the point that it is impossible to claim that mean-based estimators are uniquely the best, in the second experimental setup we consider an exponential bandit with K=8 arms. We observe that it is unclear as to which estimator provides a better sample complexity. This is expected, since exponential distribution is not heavy-tailed, although the median and mean are distinct for the setup.
> > >
> > > | $\delta$                  | 0.025   | 0.017      | 0.01   | 0.005      | 0.001      |
> > > |---------------------------|---------|------------|--------|------------|------------|
> > > | Algorithm 2: trimmed mean | 7223.4  | **7264.8** | 7865.8 | **7838.4** | **8332.3** |
> > > | Algorithm 2: median       | **7151.11** | 7433.8  | **7652.2**  | 8337.0     | 8424.5     |
> > >
> > > Finally, as the reviewer suggests, a promising future direction would be to study the CBAI in the sub-exponential setup (to address the case of heavy-tailed distributions). However, we would like to digress; while studying the problem in the sub-exponential setup is promising, that does not take away the fact that CBAI has not been studied in the sub-Gaussian setting, and thus, does not diminish the significance and novelty of the observations.
> > >
> > > **Comment:** I think it would be useful to emphasize the dependency of the complexity $H$ on $\varepsilon$, but also on the problem instance to clarify that what you have is an instance-specific lower bound that also depends on $\varepsilon$.
> > >
> > > **Response**: Thank you very much for the valuable suggestion. We will definitely highlight the relationship that the problem complexity $H$ has on $\varepsilon$ in the revised version of the paper.
> > >
> > > **Comment**: I am however a bit curious on how $U_i$ depends on the arm $i$, and whether that affects the dependence on $\varepsilon$.
> > >
> > > **Response**: Regarding how $U_i$ depends on the arm $i$, and whether that affects the dependence on $\varepsilon$ we note that the uncertainties model follows that of [4], which uses separate uncertainties $U_i$ for the partially identifiable best arm identification (PIBAI) setup, of which, CBAI is a special case. The PIBAI is more general, in the sense that it assumes that each arm can only be identified up to an uncertainty $U_i$. Indeed, for CBAI with a fixed contamination level $\varepsilon$, the $U_i$’s are the same for each arm $i$. We could readily extend the current CBAI setup to the case where we have different (known) levels of contamination $\varepsilon_i$ for different arms. In that case, each arm $i$ would have a different $U_i$ depending on the level of contamination of that arm $\varepsilon_i$.
> > >
> > > **Comment**: I believe that the authors in [4] also unveil that there is a bias that we cannot avoid when estimating the means (the U_i) and list this as one of their contributions.
> > >
> > > **Response**: Indeed, it is natural and quite expected that when the samples are contaminated, their empirical mean (or any other statistics) deviates from the true value (hence, the bias). Hence, establishing that the uncertainties (bias) do exist follows the first principles. Indeed [4] answers the natural question of “what is the level of bias?” as their contribution in this respect.  We also address the same fundamental question. Expectedly, due to the distinctions in the setup, the model, and the algorithms, the answers and their attendant analyses are distinct. Specifically, for  a more general class of models (*any* sub-Gaussian model), we quantify the bias range ($U_i = \mathcal{O}(\sigma\varepsilon\sqrt{\log(\frac{1}{\varepsilon})}))$), which matches the universal lower bound on the minimum possible deviation ($U_i = \Omega(\sigma\varepsilon\sqrt{\log(\frac{1}{\varepsilon})}))$ [28]) for sub-Gaussian distributions. In this sense, the trimmed-mean estimator is optimal under the sub-Gaussian setup.
> > >
> > > For completeness, we also provide the (perhaps trivial) reason for why the mentioned biases do exist. The reason lies in the fact that it is impossible to uniquely identify the data generation and corruption models from the mixture distributions generating the rewards in each arm. Thus, the level of uncertainty (or bias) in estimating the statistical characteristics depends on the complexity of the set of probable data generation and corruption model pairs that  may generate the same mixture model.

---

> > > > ### Author Response · Authors · 2021-08-31
> > > > **Kindly inquiring if you had a chance to examine the response**
> > > >
> > > > Given the short period remaining, we'd like to kindly inquire if you had a chance to take a look at our responses.
> > > >
> > > > We'd be grateful if you re-evaluate your rating of the paper, if you believe we've addressed your concerns. Otherwise, we'll be more than happy to further elaborate on the remaining concerns.

---

> > > > > ### Comment · Reviewer_xUqy · 2021-08-31
> > > > > **Thanks for your responses!**
> > > > >
> > > > > Thanks again for your responses. You have clearly addressed my concerns and I think that I require no further clarifications for the time being.

---

### Official Review · Reviewer_ccu8 · 2021-07-14

**Rating:** 5
**Confidence:** 5

**Summary:**

This paper studies the best-arm-identification problem in the Huber contamination model where each arm has some probability epsilon of being sampled from some adversarial distribution.

The setting in [4] was modified by replacing the median estimator with a trimmed-mean estimator. Since adversarial corruptions can make the original mean unidentifiable, the authors assume that the corruptions do not change the ranking of the arms.

The authors propose a new bandit algorithm, based on this estimator, and provide upper bounds on its regret. They also provide a new lower bound for any partially identifiable bandit instance.

The proposed algorithms are evaluated on both real and synthetic data.



**Limitations And Societal Impact:**

yes

**Main Review:**

I'll be begin by saying that the paper was a pleasure to read; the prose was clear, the results complete (in that matching upper and lower bounds were provided), and paper as a whole flowed nicely.

However, my huge reservation is that this work does not add much beyond the results from [4] aside from empirical evaluations. From what I can tell, the only difference is that the median estimator was replaced by a alpha-trimmed mean estimator which allows for correct identification of the mean when the arms are subgaussian. While the new algorithm is analyzed to a satisfactory extent, it does not add enough to our understanding of the contaminated bandit problem to warrant a separate publication. Additionally, the paper should be clearer about what is novel and what is inherited from prior work, as it occasionally reads as innovating much more than it should.

How are the assumptions on bandit instances which convergence of the median more restrictive, as claimed on line 246?

Another worry is that the experiments do not describe the corruption model; how sensitive are the algorithms to the strength of the corruptions, even in the oblivious setting?

I would also advise the authors to choose a different name, as the title is excatly shared with [4] except with an "in" replacing a "for."

post-discussion: I have increased my score to 5 as a result of my greater appreciation for the results.

**Time Spent Reviewing:**

3

---

> ### Author Response · Authors · 2021-08-10
> **Response to reviewer ccu8 of the Paper9665**
>
> **Comment**: However, my huge reservation is that this work does not add much beyond the results from [4] aside from empirical evaluations. From what I can tell, the only difference is that the median estimator was replaced by a alpha-trimmed mean estimator which allows for correct identification of the mean when the arms are subgaussian.
>
> **Response**:  We, respectively, strongly disagree with the observations of the reviewer. We provide some high-level major differences in the settings, models, and the nature of results:
>
> 1- Different Bandit Settings: Our algorithms work for *any* sub-Gaussian bandit setting. However, those in  [4] work for a limited subset of these settings, which does not include some of the important cases (e.g., Bernoulli bandit). The details of the class of models can be found in definition 2 in [4], and they are based on strict regulatory conditions on the CDF of the arms.
>
> 2- Distinct Algorithms: Another major distinction is that gap-based algorithms have not been analyzed in the CBAI setup (or in [4]); the additivity property of the suboptimality gaps required in the analysis of gap-based methods in the non-contaminated setup fails to hold in the CBAI setup. To get around this issue, we show that a fixed exploration phase of at least $\sqrt{t}$ samples per arm is necessary for showing asymptotic optimality in terms of sample complexity. Such exploration phases are unnecessary for optimality in the non-contaminated setup for gap-based algorithms.
>
> 3- Performance guarantees: Apart from these, we have analyzed two procedures and provided both theoretical and empirical evidence of their superior performance. Such insights were not present in the literature for BAI in the contaminated setup, as has been pointed out by reviewer 82x7, who has mentioned that “.. this work is well-organized and it is interesting in the sense the mean-based BAI problem with fixed-confidence setting has not been explored for the contaminated bandits before”.
>
> 4- Technical differences: there are a large number of steps in the analysis that are distinct from those in the existing literature. The key difficulty in the analysis arises from the fact that it is impossible to estimate the mean of an arm any better than an uncertainty interval $U_i$ around the true mean, even if we have an infinite number of samples from that arm [4]. This leads to two principle challenges:
>
> i) *Estimator concentration*: While the estimator concentration relies on standard inequalities for sub-Gaussian distributions,  the corrupted samples inflict a deviation $U_i$ from the true mean for each arm $i\in[K]$. This motivates the use of tools from the robust statistics literature (specifically, the resilience property of sub-Gaussian distributions) to upper bound the maximum deviation that the trimmed-mean estimator suffers due to the contamination. We observe that this deviation matches the lower bound on the error-guarantee on sub-Gaussian distributions [28]. To further reinforce the gain obtained by using the trimmed-mean estimator (instead of the sample mean, as used in optimal algorithms for BAI), we have performed more experiments with the algorithms for BAI operating on contaminated samples. Specifically, we have simulated the successive elimination algorithm and the gap-based algorithm (attack-free BAI counterparts) and compared them against the proposed Algorithm 2 and Algorithm 1 with a corruption level of $\varepsilon=0.1$. The results are as follows:
>
> | $\delta$               | 0.025         | 0.017         | 0.01          | 0.005         | 0.001         |
> |------------------------|---------------|---------------|---------------|---------------|---------------|
> | Successive Elimination | 6988.8        | 6994.5        | 7359.4        | 7491.7        | 8002.9        |
> | Algorithm 2            | **6943.4** | **6992.4** | **7128.5** | **7429.9** | **7966.9** |
>
> ---
>
> | $\delta$                | 0.025      | 0.017    | 0.01       | 0.005      | 0.001      |
> |-------------------------|------------|----------|------------|------------|------------|
> | Gap-based (attack-free) | 4328.4     | 4529.5   | 5059.5     | 5620.3     | 6959.1     |
> | Algorithm 1             | **4243.5** | **4521** | **4995.1** | **5572.8** | **6900.2** |
>
>
>
> The above results clearly demonstrate the advantage that the trimmed-mean estimator provides in the contaminated setting over the sample-mean.
>
> ii) *Sample complexity upper bound for Algorithm 1 (Theorem 4.3)*:  The analysis for the attack-free counterpart of Algorithm 1 heavily relies on the property of the additivity of suboptimality gaps [24]. Specifically, the additivity of the suboptimality gap implies that $\Delta(i,j) = \Delta_i - \Delta_j$, where we have defined $\Delta(i,j) \triangleq \mu_j - \mu_i$ and $\Delta_i \triangleq \mu_{a^\star} - \mu_i$ for any arm $i,j\in[K]$. However, it can be readily verified that this property fails to hold in the contaminated setup due to the uncertainty intervals $U_i$. To get around this difficulty, we resort to a forced exploration phase of at least $\sqrt{t}$ pulls per arm in order to ensure that we have sufficient samples to estimate the means of each arm (and hence, the best arm) within the uncertainty interval up to a sufficient level of tolerance. This ensures that $\Delta_{\hat a_t}\rightarrow\Delta_{a^\star}$ asymptotically. Such fixed exploration phases are not required for gap-based  algorithms in the attack-free setting, but it becomes a key factor in obtaining the sample complexity bound in the CBAI setting. We refer the reviewer to the proof of Theorem 4.3 and Appendix C for the details.
>
> **Comment**: How are the assumptions on bandit instances which convergence of the median more restrictive, as claimed on line 246?
>
> **Response**: As mentioned in item 1 of the previous response, the class of bandit instances studied in [4] depends on strict regularity assumptions on the cumulative distribution function (CDF) (cf. definition 2 of [4]), which does not include a wide range of important distribution models (e.g., Bernoulli bandits). In contrast, our guarantees are based on the sub-Gaussian assumption, and hence works for *any* sub-Gaussian bandit instance.
>
> **Comment**: the experiments do not describe the corruption model; how sensitive are the algorithms to the strength of the corruptions, even in the oblivious setting?
>
> **Response**: The description of the corruption model can be found in Appendix G (lines 588 through 590). The ablation study on various $\varepsilon$ values can be found in Fig. 2. We would also like to refer the reviewer to Fig. 4b, where we have used an exponential bandit setup to compare the proposed algorithms (with the theoretical confidence intervals) against [4]. The experiment shows that the proposed Algorithm 2 outperforms median-based successive elimination proposed in [4]. The sensitivity of the proposed algorithms 1 and 2 to increasing contamination level $\varepsilon$ has been plotted in Figure 2.
>
> **Comment**:  would also advise the authors to choose a different name, as the title is excatly shared with [4] except with an "in" replacing a "for."
>
> **Response**: We absolutely agree with the reviewer. We intended to have a reference to ” mean-based estimator” in the title, which was unfortunately overlooked and we would be adjusting it.

---

> > ### Comment · Area_Chair_2qXY · 2021-08-23
> > **Respond to response**
> >
> > Hi Review ccu8,
> >
> > Could you check the authors' response and see if they address your concerns? It seems there are some strong disagreements that rolling discussions can help to resolve.

---

> > ### Comment · Reviewer_ccu8 · 2021-08-26
> > **follow up**
> >
> > Thanks for your thorough response. I admit that I underestimated the contributions at first and will be increasing my score. In particular, obtaining asymptotic optimality is new in this setting.
> >
> > Is the concentration result, Lemma 4.1, new? Trimmed mean estimators are very common in the literature, so I would be a bit surprised. Also, can you evaluate the constants in (13), and wouldn't you need to in order to apply this concentration bound in a bandit algorithm?

---

> > > ### Author Response · Authors · 2021-08-27
> > > **Response to the follow-up discussion**
> > >
> > > We thank the reviewer for taking the time to evaluate the response notes and following up on the discussions. To answer the two new questions raised, we provide the following discussions.
> > >
> > > **Comment**: Is the concentration result, Lemma 4.1, new? Trimmed mean estimators are very common in the literature, so I would be a bit surprised.
> > >
> > > **Response**: To the best of our knowledge, we are unaware of any concentration result similar to Lemma 4.1. As the reviewer has correctly pointed out, indeed the trimmed mean estimator is common in the robust statistics literature, and there are several concentration results for the trimmed mean estimator in the literature, each specific to the problem setup and assumptions. To highlight the distinctions, we provide two of the existing ones that are most relevant to our scope:
> > >
> > > 1- The most relevant trimmed mean concentration (closest to our setting) can be found in Theorem 1 of [Niss2020], which considers our $\varepsilon$-contamination model and derives an estimator concentration for the purpose of regret minimization in the contaminated setting. However, the paper does not consider the uncertainty $U_i$ in estimating the arm means in the contaminated setting, and thus provides a concentration result which is much weaker compared to ours for large $t$ (owing to the $\sqrt{\log t}$ term in the concentration bound, which may increase massively for large $t$). In this sense, our concentration result is considerably tighter.
> > >
> > > 2- As for a second example, consider Lemma 1 of [Bubeck2012]. In the setting of the paper, the underlying assumption does not consider a contamination model like we consider, and are derived for the class of heavy-tailed distributions (with moments of the order of $1+\varepsilon$, $\varepsilon\in(0,1)$). Thus, the concentration result does not take into account the uncertainty $U_i$ as a result of the corrupted samples. This is an example of a trimmed mean concentration studied in a different setting.
> > >
> > > **Comment**: Also, can you evaluate the constants in (13), and wouldn't you need to in order to apply this concentration bound in a bandit algorithm?
> > >
> > > **Response**: Thanks for the thoughtful question. we **do not** need to know the constant term for the proposed bandit algorithms. The reason for this is the modified guarantee on the detection confidence $\delta$ (Definition 2.2), and the Assumption (i) which requires the arm means to be more than the uncertainty levels $U_i = \mathcal{O}\left(\sigma\varepsilon\sqrt{\log\frac{1}{\varepsilon}}\right)$ apart, so that the detection task is feasible.
> > >
> > >
> > > We also note that we can evaluate the constants hidden in the complexity notation for the bias in the estimator concentration (13). Specifically, let us denote the constant by $c$, in which case we have that any $c\geq4+\sqrt{2}$ satisfies (13), which is a consequence of equations (33), (43) and (44) in Appendix B.
> > >
> > > [Niss2020] Niss, L. &amp; Tewari, A.. (2020).  What You See May Not Be What You Get: UCB Bandit Algorithms Robust to $\varepsilon$-Contamination. *Proceedings of the 36th Conference on Uncertainty in Artificial Intelligence (UAI)*
> > >
> > > [Bubeck2012] Bubeck, S. &amp; Cesa-Bianchi, N. &amp; Lugosi, G.. (2012) Bandits with heavy tail. *arXiv1209.1727*
> > >
> > > **Finally**, we would like to, kindly, ask the reviewer to re-evaluate their rating of the paper if all their questions have been addressed. We would like to **emphasize one high-level discussion pertinent to the significance and novelty** of the results:
> > >
> > > Our paper is the first to provide algorithms with provable guarantees for the **standard** BAI problem under the **standard** data models (sub-Gaussian) with contamination. Please note that the only relevant study is that of [4] which has two significant differences:
> > >
> > > 1- [4] considers a **modified** BAI problem (looking for the best **median** instead of the best **mean**) distinct from that the canonical definition of the BAI problem.
> > >
> > > 2- It focuses on a a significantly more restricted class of models (specified in [4, Definition 2]), which does not include many popular and widely-analyzed classes of bandit instances (e.g., heavy-tailed continuous models and all discrete models such as the Bernoulli). For convenience in checking these, we are copying Definition 2 of [4]:
> > >
> > > (Definition 2, [4]): *For any $\bar{t}\in\bigg(0,\frac{1}{2}\bigg)$ and  $B>0$, let $F_{\bar{t},B}$ be the family of distributions $F$ that have a unique median and satisfy $\lvert F(x_1) - F(x_2)\rvert \geq \frac{1}{Bm_2(F)}\lvert x_1 - x_2\rvert$ for  all $x_1,x_2\in[Q_{L,F}(\frac{1}{2}-\bar{t}),Q_{R,F}(\frac{1}{2}+\bar{t})]$.*
> > >
> > >
> > > In the above definition, $m_2(F)$ is the median absolute deviation. The definition essentially refers to the class of cumulative distribution functions (CDFs) that are *not too flat* near the median, and increase at least linearly. We are the first to study the contaminated **standard** best arm identification (CBAI) problem in the **sub-Gaussian** setup.

---

### Decision · Program_Chairs · 2021-09-27

**Decision:**

Accept (Poster)

**Comment:**

Overall, reviewers are mostly positive about this paper (especially after author response). The reviewers agree that the paper studies a new and relevant problem, and the concentration bound result is interesting. According to reviewers' comments and discussions, it would be nicer if authors can improve the presentation to highlight the challenges and improvement over prior work.